# Do Chemical-Based Bonding Techniques Affect the Bond Strength Stability to Cubic Zirconia?

**DOI:** 10.3390/ma14143920

**Published:** 2021-07-14

**Authors:** Allegra Comba, Andrea Baldi, Riccardo Michelotto Tempesta, Massimo Carossa, Letizia Perrone, Carlo Massimo Saratti, Giovanni Tommaso Rocca, Rossella Femiano, Felice Femiano, Nicola Scotti

**Affiliations:** 1Department of Surgical Sciences, CIR Dental School, University of Turin, Via Nizza 230, 10126 Turin, Italy; allegra.comba@unito.it (A.C.); andrea.baldi@unito.it (A.B.); riccardo.tempesta@hotmail.it (R.M.T.); massimo.carossa@unito.it (M.C.); letizia.perrone@edu.unito.it (L.P.); 2Division of Cariology and Endodontology, School of Dentistry, University of Geneva, 1211 Geneva, Switzerland; saratti.carlomassimo@gmail.com (C.M.S.); giovanni.rocca@unige.ch (G.T.R.); 3Restorative Dentistry, University of Study of Campania, 81100 Naples, Italy; rossella.femiano@libero.it (R.F.); femiano@libero.it (F.F.)

**Keywords:** cubic zirconia, 10-MDP, tribochemical adhesion, shear-bond strength, thermocycling

## Abstract

This study evaluated the effectiveness of chemical-based adhesive techniques on promoting immediate and aged bond strength between zirconia and luting cement. A total of 128 discs of zirconia were divided into 4 groups (*n* = 32) according to the adhesive treatment: tribochemical silica-coating followed by silane (Silane Primer, Kerr) and bonding (Optibond FL, Kerr), Signum Zirconia Bond (Hereaus), Z-Prime Plus (Bisco), and All-Bond Universal (Bisco). Composite cylinders were cemented on the zirconia sample with Duo-Link Universal (Bisco). Eight specimens per group were subjected to 10,000 thermocycles and subsequently bond strength was tested with shear-bond strength test. ANOVA test showed that artificial aging significantly affected the bond strength to zirconia. Bonferroni test highlighted a significant influence of adhesive treatment (Signum) on bond strength after thermocycling. It was concluded that 10-MDP-based bonding systems showed no improvement in initial bond strength compared with tribochemical treatment. All chemical bonding techniques tested in this study were influenced by thermocycling.

## 1. Introduction

Continued development of dental ceramics has allowed the coupling of excellent esthetic qualities with improved mechanical properties, and these ceramics are now widely used to realize indirect metal-free restorations. Their major properties are hardness, a high coefficient of elasticity, resistance to heat and chemical attack, and high fragility [1,2,3]. Several clinical studies of all-ceramic restorations have shown that fracture is the main reason for failure [4]. In fact, ceramics are brittle and stiff materials, showing elastic modulus between 65 GPa (feldspathic and glass ceramics) and 250 GPa (alumina- and zirconia-based ceramics) [5]. Moreover, glass ceramics show flexural strengths comparable to resin composites (70–130 MPa), whereas pure crystalline ceramics have a higher flexural strength, around 900 MPa [6,7]. Today, with advances in CAD/CAM technology, it has become possible to realize restorations with improved strength, marginal fit, and esthetics with materials that could not otherwise be managed [8].

Dental ceramics could be classified according to the silica content. The silica-based ceramics (SiO_2_), which include feldspathic, leucite-based, and lithium-disilicate materials, have well-established adhesive protocols due to the presence of silica. In contrast, alumina and zirconia, also called crystalline ceramics, do not contain silica or glass components. Their atoms are arranged in a regular pattern, which makes them denser and more resistant to fracture [9,10], but they have poorer esthetic properties [11]. Cubic zirconia has been recently introduced in restorative dentistry, replacing the tetragonal version, especially for monolithic single-tooth restorations. The introduction of a variable amount of cubic phase, which is optically isotropic, was meant to improve the translucency of the material, at the expense of strength and toughness due to the lack of transformation toughening and the coarser microstructure [12].

Independently of the mechanical properties, the use of cubic zirconia over non-retentive preparations is still challenging, due to the difficulties in obtaining a predictable adhesion over it. The absence of a glassy phase makes the bonding mechanisms to dental tissues more difficult [13,14]. To overcome the absence of silica, a silicatization procedure was suggested in several studies [15,16,17]. However, different adhesive protocols have been proposed to process and pretreat the crystalline ceramic surface to promote adhesion between the ceramic and resin cement [18]. A recent review by Ozcan and Bernasconi [19] analyzed the adhesion mechanisms of resin-based and glass-ionomer luting cements to zirconia and sought to determine factors affecting bond strength. The authors investigated 169 different surface-conditioning methods, mainly combinations of air-abrasion protocols and adhesive promoters, such as primers or silanes. They concluded that adhesion of the luting cements was influenced significantly by the surface-conditioning method, cement type, test method, and aging condition. The results indicated that physico-chemical conditioning methods tended to increase the bond strength values for resin-based cements.

To date, researchers have focused on chemical interactions between adhesive systems and zirconia to enhance bond strength and durability. A well-known chemical conditioning method to promote adhesion is based on the 10-methacryloyloxydecyl dihydrogen phosphate (10-MDP) monomer. The 10-MDP monomer has two functional groups, an acid group that bonds to hydroxyl groups on the zirconia surface and a methacrylate (carboxylic acid group) that can be light-cured with the composite monomers. The 10-MDP monomer is contained in specific primers designed for zirconia [20], but recently, it has been added in several universal adhesives, which are single-bottle adhesives that can be used on different substrates [21]. The manufacturers claim that these adhesives can promote bonding of methacrylate-based materials to various indirect restorative materials. In fact, universal adhesives present functional monomers in their composition to promote adhesion to surfaces based on metallic oxides, such as zirconia and other metal alloys. Recent literature showed different and non-standardized aging methods to test the stability of the bonding between 10-MDP and zirconia, which allow a great variability in the obtained results [22,23]. Thus, the aim of this in vitro study was to evaluate the effectiveness of different chemical-based adhesive techniques in promoting bond strength between zirconia and luting cement. The null hypotheses were that (I) the adhesive technique does not influence the initial bond strength and (II) the bond strength is not affected by artificial aging.

## 2. Materials and Methods

### 2.1. Study Design

This study was designed in 8 study groups (*n* = 16 each), where the specimens were randomly allocated (https://www.randomizer.org/, accessed on 17 June 2021) considering:

“Adhesive Protocol” in 4 levels: four different adhesive protocols for cubic zirconia bonding, were selected (32 specimens each): tribochemical treatment, Signum Zirconia Bond (Kulzer, Hanau, Germany); Z-Prime Plus (Bisco, Schaumburg, IL, USA); All-Bond Universal (Bisco, Schaumburg, IL USA).

“Artificial aging” in 2 levels: half of the specimens treated with an adhesive approach were thermocycled before shear-bond stress.

A schematic representation of the study design is displayed in Figure 1.

### 2.2. Specimens Preparation

For this in vitro study, 128 square-shaped specimens 2 mm thick were prepared by sectioning CAD-CAM blocks of cubic zirconia with a diamond saw (Katana STML, shade A2, Kuraray Noritake, New York, NY, USA). After sectioning, all specimens were sintered and glazed according to the manufacturer instructions. The obtained slices were polished with #600, #1000 and #1200 paper grit to standardize zirconia surface.

### 2.3. Shear-Bond Strength Test

The so-obtained specimens were embedded in an acrylic resin base (Resin LC, Henry Shein, Germany), repolished with paper grit to remove any residual resin that might have covered the zirconia surface. Specimens were the washed under copious deionized water and divided into 4 groups (*n* = 32 per group) according to the adhesive treatment performed. Before adhesive application, all the ceramic surfaces were cleaned and dried.

Group 1: Airborne-particle abrasion was performed over the ceramic surface with 30 μm tribochemical silica-coated alumina particles (CoJet sand, 3M ESPE) at 3 bar pressure for 15 s at 10 mm distance using an intraoral microblasting unit (Kavo). After washing the specimen surface for 30 s with water, one coat of Silane Primer (3M ESPE) was applied and air-dried for 10 s before brushing the bonding resin (Optibond FL, Kerr, Orange, CA, USA) over the entire surface and light-cured for 20 s with a LED curing unit at 1400 mW/cm^2^.

Group 2: Signum Zirconia Bond I (Kulzer, Germany) was dispensed and applied with a microbrush to the entire surface for 20 s and air-dried for five seconds; Signum Zirconia Bond II was then applied, and light-cured as described in group 1.

Group 3: The ceramic surface was coated with a single layer of Z-Prime Plus (Bisco Dental, Schaumburg, IL, USA), which was applied through a microbrush for 1 min and gently air-dried.

Group 4: A universal adhesive system (All-Bond Universal, Bisco) was brushed over the zirconia surface in two consecutive layers, 15 s each. After gentle air-drying, specimens were light-cured as described in group 1.

A summary of the adhesives procedures in the different groups is displayed in Table 1.

Preformed polyethylene molds with a hole in the center were then used to produce composite cylinders (diameter 2 mm, height 5 mm). The nanohybrid composite resin (Adonis, Sweden & Martina, Italy) was placed into the mold and incrementally condensed to fill it; each cylinder was cured with a multi-LED lamp (Valo, Ultradent, South Jordan, UT, USA) at 1400 mW/cm^2^ for 120 s. Then, the polyethylene mold was removed, and the cylinders were transferred to distilled water and stored in a dark box for 7 days at 37 °C. Then, after the various adhesive procedures were performed on the zirconia surfaces, the cylinders were cemented with a constant pressure on the center of the zirconia sample surface with Duo-Link Universal (Bisco), and light-cured after one minute on four sides for 20 s per side.

Next, the specimens were stored in deionized water for 24 h at 37 °C. After storage 16 specimens from each group were immediately subjected to the shear-bond strength test, while the remaining 16 specimens of each group were subjected to 10,000 thermal cycles in alternating water baths at 5 °C and 55 °C for 60 s each (5 s dwell time). The shear-bond test was performed fixing the resin base on a universal machine (Ultradent Products Inc., South Jordan, UT, USA) with a notched blade that applied a force on the composite cylinder at a speed of 1 mm/min. This test provides bond strength expressed in MPa/mm^2^, which is calculated using the formula σ = L/A, where L is the force (expressed in N) at which there was the detachment of the cylinder, A = π × r^2^ (in mm^2^) was the bonding surface, and r is the radius of the cylinder.

Fractured specimens were observed under optical microscopy (Wild, Heerbrugg, Gaiss, Switzerland) at 40× magnification to establish failure modes, which were classified as: ACO (adhesive detachment between composite and resin cement); ACE (adhesive detachment between ceramic and resin cement); AM (mixed adhesive detachment); CC (cohesive detachment within composite) and CM (cohesive detachment within ceramic).

### 2.4. Statistical Analysis

Two-way ANOVA and Tukey *post hoc* tests were performed to evaluate the effect of different adhesion technique (AT) and artificial aging (AA), and their interaction, on bond strength. Fracture modality was analyzed using the χ^2^ test. Differences were considered statistically significant at *p* < 0.05.

## 3. Results

Mean bond strengths (with standard deviations) and type of fracture (expressed as percentage) in each group before and after thermocycling are shown in Table 2. During thermocycling, spontaneous debonding of composite was observed: one from group 1, three from group 2, and two from group 4.

The ANOVA test showed that adhesive treatment (*p* < 0.00001) and the artificial aging (*p* < 0.0001) significantly affected the bond strength on zirconia. The Tukey analysis highlighted a significant influence of adhesive treatment on bond strength after thermocycling. In particular, the bond strength obtained in group 2 was significantly higher than that for other adhesive procedures (*p* < 0.0001). In regard of the fracture mode, ACO was the mainly detected fracture in all groups, before and after thermocycling. The χ^2^ test revealed that AT and AA did not statistically influence the type of fracture between zirconia and dual-curing cement.

## 4. Discussion

Based on the obtained results, the first null hypothesis was accepted since the adhesive techniques tested did not significantly influenced the initial bond strength on cubic zirconia.

In this in vitro study, the shear-bond strength test was used to evaluate the bond strength of resin cement to zirconia after treatment with universal adhesive or specific zirconia primers. Currently, methods used to evaluate bond strength include the shear-bond strength test [17,24,25], the micro-shear-bond strength test [26], and the microtensile bond strength test [27,28]. Some authors prefer the microtensile bond strength test because it allows a more homogenous distribution of stresses, but it has been criticized because of difficulties in sectioning bonded zirconia into microbeams without damaging the adhesive interface [29] and because of the high number of premature failures during sample sectioning [30,31]. The shear-bond strength test is a relatively simple and quick method, and it has been used in several studies to evaluate the bond strength on ceramic materials [24,32]. This test can, however, develop inhomogeneous forces at the interface, which could promote cohesive fractures in the substrate [31,33]. However, in the present study the fracture analysis showed, above all, adhesive fractures between the zirconia and cement.

The adhesion between specific primers or universal adhesives and zirconia is based on the interaction of organophosphates monomers (10-MDP) with the hydroxyl groups on the zirconia surface. A recent in vitro study [34] confirmed that MDP can establish a chemical bond with tetragonal as well as cubic zirconia, above all if applied on an alkaline pH surface. Inokoshi and Van Meerbeek [13] also affirmed that the use of a 10-MDP-containing primer revealed a higher predictable bond strength than when zirconia was treated with another primer or received no chemical pretreatment. In their meta-analysis the highest predictable bond strength was found when zirconia was tribochemically silica-coated and additionally chemically treated with an MDP-containing primer.

Regarding the effectiveness of chemical bonding techniques, the results of the present study are consistent with those reported by Seabra et al. [35], who concluded that new multimode MDP-containing adhesives can be used effectively to promote adhesion between composite resin and zirconia, although without obtaining any improvement in immediate bond strength compared with ceramic primers. In fact, the authors found no statistical differences in the bond strength of the specimens treated with universal adhesives (Scotchbond Universal, 3M ESPE, and All-Bond Universal, Bisco) and a specific primer for zirconia (Z-Prime Plus, Bisco), which was confirmed by the results obtained in the present work. Other previous studies evaluated the bond strength on zirconia obtained with specific primers and/or universal adhesives. Ural et al. [20] tested the effect of different primers on the bond strength of adhesive resin cement to a zirconia surface using three different primers: Monobond S (Ivoclar Vivadent), ClearFil Ceramic Primer (Kuraray Dental), and Signum Zirconia Bond (Heraeus Kulzer). Their results agreed partially with the present study in that the highest shear-bond strength values were obtained with Signum Zirconia Bond, although this finding did not apply bonding under conditions of artificial aging procedures. Piascik et al. [32] evaluated the contact angle and shear-bond strength of two zirconia primers, one universal adhesive, and a recently developed fluorination pretreatment. The results revealed that MonoBond Plus and ClearFil Ceramic Primer had statistically similar low bond strengths, whereas the plasma fluorination group and the Z-Prime group displayed the highest bond strengths. MonoBond Plus (Ivoclar) is a universal adhesive, and ClearFil Ceramic Primer (Kuraray) is a primer for any type of ceramic and composite. Both materials contain silane and phosphate components and showed contact angles similar to that of native zirconia, suggesting limited chemical bonding to the surface, whereas Z-Prime contained phosphate and carboxylic monomers and thus showed a lower contact angle and a stronger chemical adhesion to cubic zirconia.

The tribochemical treatment consists of surface roughening with 30-µm silica modified with Al_2_O_3_ particles, followed by silane application. The tribochemical treatment allows combining micro-mechanical retention created by sandblasting with the chemical retention achieved by silanization. In the present study, the tribochemical treatment did not increase the immediate bond strength of the resin cement to zirconia compared with ceramic primers and universal adhesives, and it was more susceptible to damage by thermocycling then some chemical bonding treatments. Several previous studies investigated the bond strength obtained between a silanized zirconia surface and resin-based cement, but there is still no consensus in the literature. Passos et al. [36] evaluated the adhesive quality of four different resin cements to zirconia; they concluded that tribochemical silica-coating promoted higher and more stable bond strength, independent of the resin cement used. In contrast, Kern and Wegner [37] claimed that tribochemical silica-coating of zirconia did not result in a durable resin bond, as it did with glass-infiltrated alumina ceramic. Moreover, according to some reports [38,39], surface roughening can cause cracks in the zirconia, which could reduce its long-term strength. This technique is also a time-consuming procedure, and its efficacy is related to the characteristics of zirconia: high-purity zirconia has high hardness and density, which may impede sufficient penetration of the silica particles [40]. Thus, the effectiveness of the tribochemical treatment could depends on the penetration of the surface of the silica-coated particles.

To better assess the long-term effectiveness of the adhesive treatments, samples were subjected to a thermocycling procedure entailing alternating baths of distilled water at 5 °C and 55 °C for 60 s each. In the present study, 10,000 thermal cycles were performed, after which some samples showed spontaneous debonding of the cylinders and they were consequently excluded from the study. Six cylinders had broken away: one from group 1, three from group 2, and two from group 4. Thermocycling is generally used to cause rapid aging of a composite; switching between high- and low-temperature baths produces expansion and contraction of the composite, which places stress on the adhesive interface, simulating the aging conditions of the mouth. Thus, thermocycling induces the hydrolytic degradation of the bonding because of the water diffusion into the interfacial layer of the resin composite and zirconia, decreasing the bond strength of resin luting agents to zirconia [17,41,42]. Regarding aging conditions, Ozcan and Bernasconi [19] affirmed that aging via thermocycling for at least 5000 cycles should be adopted to test stability of bonding to zirconia. In the present study, 10,000 cycles were performed to simulate on year of in vivo functioning.

The results of the present study caused us to reject the second null hypothesis because thermocycling significantly influenced the bond strength for all adhesive techniques tested in this in vitro study. These results were partially supported by a previous study by Kim et al. [43], which evaluated the bond strength of universal adhesives to zirconia before and after thermocycling. They concluded that the bond strength for all conditioning agents were reduced significantly after thermocycling, consistent with the present study. Additionally, in Kim’s study, All-Bond Universal showed significantly higher bond strength compared with Single Bond Universal and Alloy Primer after thermocycling, which differs from our results. Unlike the primers tested in the present in vitro study, the primer tested by Kim et al. contained MDP but did not contain other resin adhesive components, which can copolymerize with the resin cement, increasing the bond strength.

In the present study the samples treated with specific primers for zirconia (Z-Prime Plus and Signum Zirconia Bond) showed similar bond strengths to other techniques. However, after thermocycling, Signum Zirconia Bond maintained significantly higher bond strength than other treatments. This outcome may be related to the primer composition. Signum Zirconia contained a mixture of organophosphate monomers that bind chemically to the zirconia on one side and to the composite resin on the other side. In particular, organophosphates monomers have an organofunctional component, often a methacrylate group, which can be co-polymerized with the monomers of the resin composite, and a phosphoric acid that binds with the metal oxides of zirconia. The other monomers cooperate in the development of the bond between the zirconia and resin cement. Moreover, the presence of acetone in Signum Zirconia Bond could increase the surface wettability of zirconia and thus promote bonding with methyl–methacrylate. Acetone can also remove impurities from the zirconia surface [44], thus increasing the contact area between luting cements and zirconia. Thus, even if simplified adhesives systems could reduce the operator-dependent variable on the adhesion outcome [45], it seems that a multi-step approach with the use of specific primers could benefit in obtaining a more stable bond strength over cubic zirconia.

The results obtained confirmed that there are currently adhesive strategies to be applied in the cementation of cubic zirconia. However, there is still a problem of stability of the adhesion values obtained, which prove to be susceptible to stresses such as thermocycling. Furthermore, the effect of the mechanical stresses on adhesion on zirconia should be evaluated in addition to thermal stress.

## 5. Conclusions

Within the limitations of this in vitro study, it could be affirmed that all adhesive treatments tested showed significant reduction in the bond strength after thermocycling. Moreover, the use of specific zirconia primers such as Signum Zirconia Bond showed a significantly higher bond strength than that of other adhesive procedures after thermocycling. All-Bond Universal did not achieve any improvement in bond strength compared with tribochemical treatment or other specific primers.

## Figures and Tables

**Figure 1 materials-14-03920-f001:**
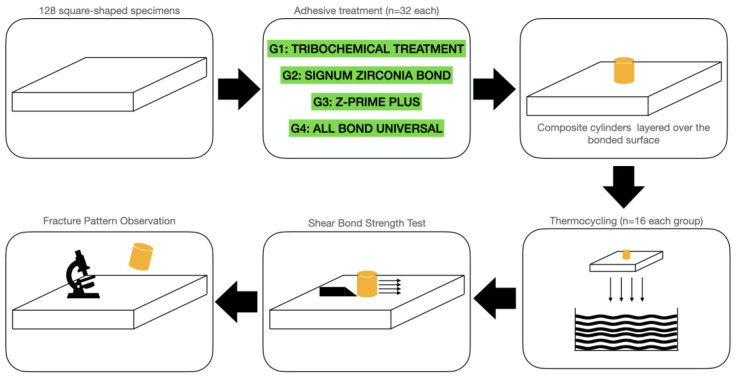
Study design.

**Table 1 materials-14-03920-t001:** Adhesive procedures performed on samples of different groups.

Adesive Protocols	Material	Type of Material	Manufacturer	Application Procedure
Tribochemical treatment (GROUP 1)	Cojet Sand	Tribochemical silica-coating particles, 30 µm	3M, Seefeld, Germany	1. air-abrasion with silica particles for 10 s. 2. wash for 30 s
	Silane Primer	silane	3M	3. apply 1 coat 4. air-dry for 10 s
	Optibond FL, Bonding	ethyl alcohol, dimethacrylate monomers, barium alumino-borosilicate glass, fumed silica, sodium hexafluorosilicate	Kerr Corp., Orange, CA, USA	5. apply 1 coat 6. gently air-dry 7. light-cure for 20 s.
Signum (GROUP 2)	Signum Zirconia Bond	Bond I: acetone, 10-mdp, acetic acid Bond II: Methyl–methacrylate, diphenyl(2,4,6-trimethylbenzoyl) phosphinoxide	Kulzer, Hanau, Germany	1. apply 1 coat of Bond I 2. air-dry for 10 s 3. apply 1 coat of Bond II 4. air-dry for 10 s 5. light-cure for 20 s
Z-Prime (GROUP 3)	Z-Prime Plus	Organophosphate monomer (MDP), carboxylic acid monomer (BPDM), HEMA, ethanol	Bisco, Inc., Schaumburg, IL, USA	1. apply 1 coat 2. air-dry for 10 s
Universal Adhesive (GROUP 4)	All-Bond Universal	Organophosphate monomer (MDP), BisGMA, HEMA, ethanol, water, initiators	Bisco, Inc., Schaumburg, IL, USA	1. apply 1 coat 2. air-dry for 10 s 3. light-cure for 20 s

**Table 2 materials-14-03920-t002:** Mean bond strength results (± SD), expressed in MPa, and type of fracture, expressed as percentage, obtained in the different groups. Different superscript uppercase letters indicate significant differences between data within the same column (*p* < 0.05). Different subscript lowercase letters indicate significant differences between data within the same row (*p* < 0.05).

Study Groups	Not Thermocycled		Thermocycled	
	Bond Strength (MPa) ± SD	Type of Fracture	Bond Strength (MPa) ± SD	Type of Fracture
Group 1	21.99 ^A^_a_ ± 4.98	16.67% ACE 83.33% ACO	8.20 ^C^_b_ ± 1.86	18.75% ACE 75% ACO 6.25% AM
Group 2	25.21 ^A^_a_ ± 5.45	100% ACO	14.53 ^B^_b_ ± 4.23	91.67% ACO 8.33% AM
Group 3	25.73 ^A^_a_ ± 5.05	100% ACO	9.31 ^C^_b_ ± 4.31	100% ACO
Group 4	23.12 ^A^_a_ ± 5.14	100% ACO	8.08 ^C^_b_ ± 1.45	83.33% ACO 16.67% AM

## Data Availability

The data presented in this study are available on request from the corresponding authors.

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
