# Peer review of "Do Chemical-Based Bonding Techniques Affect the Bond Strength Stability to Cubic Zirconia?"

_materials, 2021, doi:10.3390/ma14143920_

Round 1
Reviewer 1 Report
dear Authors, first of all, I suggest to revise the numeration of your affiliation: use your initials and reduce it.
The research is well designed and carried out.
Abstract: it is a good summary of the paper, and it it well organized.
Introduction contains enough background informations regarding the techniques involved and adequate references. The aim is well stated.
Materials and methods are clearly described, but you should consider to add some figures, in order to attract the readers, show your work and create a schematic summary.
Results: statistical analysis is carried out in a proper way, have you done the power analysis? The samples seems to involve few participants.
I would appreciate if you state the drawback of your work in discussion.
Please add some other clinical consideration at the end of discussion, you could emphasize the clinical aspect and perspectives of your work.
Conclusions are ok, but please write it as paragraph and not such as bullet points.
Author Response
Materials and methods are clearly described, but you should consider to add some figures, in order to attract the readers, show your work and create a schematic summary.
Our response: thanks for the comment, the graphical representation of the study design was added.
Results: statistical analysis is carried out in a proper way, have you done the power analysis? The samples seems to involve few participants.
Our response: thanks for the comment. For each group, 16 specimens were prepared to test bond strength through shear test. Similar studies previously published tested bond strength with similar or eventually less specimens per adhesive treatment:
Steiner R, Heiss-Kisielewsky I, Schwarz V, Schnabl D, Dumfahrt H, Laimer J, Steinmassl O, Steinmassl PA. Zirconia Primers Improve the Shear Bond Strength of Dental Zirconia. J Prosthodont. 2020 Jan;29(1):62-68. doi: 10.1111/jopr.13013. Epub 2019 Jan 22. PMID: 30624832.
Stefani A, Brito RB Jr, Kina S, Andrade OS, Ambrosano GM, Carvalho AA, Giannini M. Bond Strength of Resin Cements to Zirconia Ceramic Using Adhesive Primers. J Prosthodont. 2016 Jul;25(5):380-5. doi: 10.1111/jopr.12334. Epub 2015 Sep 15. PMID: 26372458.
Sadid-Zadeh R, Strazzella A, Li R, Makwoka S. Effect of zirconia etching solution on the shear bond strength between zirconia and resin cement. J Prosthet Dent. 2020 Nov 5:S0022-3913(20)30497-2. doi: 10.1016/j.prosdent.2020.09.016. Epub ahead of print. PMID: 33162113.
I would appreciate if you state the drawback of your work in discussion.
Our response: thanks for the comment, the text was modified accordingly.
Please add some other clinical consideration at the end of discussion, you could emphasize the clinical aspect and perspectives of your work.
Our response: thanks for the comment, the text was modified accordingly.
Conclusions are ok, but please write it as paragraph and not such as bullet points.
Our response: thanks for the comment, conclusions were modified accordingly.
Reviewer 2 Report
Presented article “ Do chemical-based bonding techniques affect the bond strength stability to cubic zirconia?” reveals the differences between different bonding techniques applied for zirconia and can be very helpful for clinicians working with full ceramic restorations. However there are some issues to be addressed:
Lines 78-81 – it would be beneficial to add couple brand names of 10-MDP monomer containing universal adhesives
Lines 83-87 - please add bibliography
Line 101 – should it be “2 levels”?
Lines 101 – 102 – do you mean that 4 group with 4 different adhesive protocols were artificially aged and 4 other groups with 4 different adhesive protocols were not subjected to artificial aging? It is not clear.
Line 104 – maybe “specimens” would be better than “disks”. Disks are usually round.
Line 113 – why n=32 per group? In line 96 you are writing about 8 groups containing 15 specimens each.
Why zirconia was not cemented to the dentine but to the composite material?
Line 145 – why shear bond strength test was performed after 24 hours storage in water and not directly after bonding?
Line 174 – should there by “adhesive” not “Adhesive”?
Line 179 – please correct the sentence “either before either after thermocycling”
Line 259 – information about spontaneous debonding should be added to the results section.
Author Response
Lines 78-81 – it would be beneficial to add couple brand names of 10-MDP monomer containing universal adhesives
Our response: thanks for the comment. A great number of universal adhesives contains 10-MDP, and adding just some brand names does not give any additional information to the scientific background of the study.
Lines 83-87 - please add bibliography
Line 101 – should it be “2 levels”?
Our response: thanks for the comment, the text was modified accordingly.
Lines 101 – 102 – do you mean that 4 group with 4 different adhesive protocols were artificially aged and 4 other groups with 4 different adhesive protocols were not subjected to artificial aging? It is not clear.
Our response: thanks for the comment, the text was modified to better explain the study design and a graphical explanation was added.
Line 104 – maybe “specimens” would be better than “disks”. Disks are usually round.
Our response: thanks for the comment, the text was modified accordingly.
Line 113 – why n=32 per group? In line 96 you are writing about 8 groups containing 15 specimens each.
Our response: thanks for the comment. Each group has 16 specimens for a total of 128. The study design was corrected.
Why zirconia was not cemented to the dentine but to the composite material?
Our response: thanks for the comment, the study aim was to test the adhesion towards cubic zirconia with different adhesives and not towards dentin or enamel.
Line 145 – why shear bond strength test was performed after 24 hours storage in water and not directly after bonding?
Our response: thanks for the comment. It is always better to wait 24 h in order to standardize the adhesive system and composite curing process, which continues for at least 12h-24h with the so-called post-polimerization process..
Line 174 – should there by “adhesive” not “Adhesive”?
Our response: thanks for the comment, the text was modified accordingly.
Line 179 – please correct the sentence “either before either after thermocycling”
Our response: thanks for the comment, the text was modified accordingly.
Line 259 – information about spontaneous debonding should be added to the results section.
Our response: thanks for the comment, the text was modified accordingly.
Reviewer 3 Report
According to the text of the manuscript, the study aimed to “evaluate the effectiveness of different chemical-based adhesive techniques in promoting bond strength between zirconia and luting cement” (lines 87-88). The first null hypothesis was that adhesive techniques tested did not significantly influence the initial bond strength of dual-curing cement on cubic zirconia (lines 185-186).
With that, according to the Materials and Methods section, “the nanohybrid composite resin (Adonis, Sweden & Martina, Italy) was placed into the mold and incrementally condensed)”. That material is not indicated for the luting purposes, this is an esthetic composite material.
Thus, it unclear, what was the aim of the study and if the authors were talking about dual-curing types of cement, and then in the Discussion of the results it is necessary to describe the results obtained from this point of view.
Materials and methods.
This study was designed in 8 study groups (n= 15) (lines 96). With that, in 2.2 Shear bond strength test the authors write about 4 groups. It is unclear. Please revise and make the Figure (Sheme) of the experiment.
Author Response
According to the text of the manuscript, the study aimed to “evaluate the effectiveness of different chemical-based adhesive techniques in promoting bond strength between zirconia and luting cement” (lines 87-88). The first null hypothesis was that adhesive techniques tested did not significantly influence the initial bond strength of dual-curing cement on cubic zirconia (lines 185-186).
With that, according to the Materials and Methods section, “the nanohybrid composite resin (Adonis, Sweden & Martina, Italy) was placed into the mold and incrementally condensed)”. That material is not indicated for the luting purposes, this is an esthetic composite material.
Our response: thanks for the comment. Even if cubic zirconia is generally luted with adhesive systems and dual-curing cement, in the present paper specimens to perform the shear bond test were prepared with a nanohybrid composite. The aim of the study was to compare different adhesive approaches, whose bond strength is not related to the composite employed for specimens preparation. However, the null hypothesis was corrected accordingly.
Thus, it unclear, what was the aim of the study and if the authors were talking about dual-curing types of cement, and then in the Discussion of the results it is necessary to describe the results obtained from this point of view.
Our response: thanks for the comment. As previously mentioned, the study aim was focused on adhesive approaches and not dual-curing cement. The text was accordingly modified.
Materials and methods.
This study was designed in 8 study groups (n= 15) (lines 96). With that, in 2.2 Shear bond strength test the authors write about 4 groups. It is unclear. Please revise and make the Figure (Sheme) of the experiment.
Our response: thanks for the comment. A schematic representation of the study design was performed and the text was modified accordingly.
Round 2
Reviewer 3 Report
Dear authors, thank you for the work done!
The necessary corrections were done, the figure describing the methodological approach was added to help to better understand the research idea. The null hypothesis was re-written and now matches all other parts of the manuscript.